# Social Interaction and Online Learning Efficiency for Middle School Students: The Mediating Role of Social Presence and Learning Engagement

**DOI:** 10.3390/bs14100896

**Published:** 2024-10-03

**Authors:** Fangfang Gao, Chunzhen Wang, Han Xie, Jianzhong Hong

**Affiliations:** 1Key Laboratory of Adolescent Cyberpsychology and Behavior (CCNU), Ministry of Education, Wuhan 430079, China; 12014538@czc.edu.cn (F.G.); 18763896501@163.com (C.W.); 2School of Psychology, Central China Normal University, Wuhan 430079, China; 3Department of Education, Changzhi University, Changzhi 046011, China; 4Institute of Educational Science, Hubei University of Education, Wuhan 430205, China

**Keywords:** social interaction, learning efficiency, social presence, learning engagement, online learning

## Abstract

(1) learning efficiency is recognized as the ultimate goal of online education, as it is related to the quality of online education and the cognitive development of students and is influenced by social interactions. This study explores the mediating roles of social presence and learning engagement in the relationship between social interaction and online learning efficiency, addressing gaps in prior studies that have not yet identified the underlying mechanisms. (2) students from three middle schools (N = 344; M_age_ = 13.61; 56.68% women) completed self-report questionnaires regarding social interaction, social presence, learning engagement, and learning efficiency. (3) the study findings reveal significant serial mediation effects of social presence and learning engagement in the relationship between learner–instructor and learner–learner interactions and learning efficiency. Specifically, while the indirect effect of learner–instructor interaction through social presence alone (indirect effect = 0.08, 95% CI = [−0.00, 0.17]) was not significant, the pathways through learning engagement (indirect effect = 0.18, 95% CI = [0.11, 0.26]) and the combined mediation through both social presence and learning engagement (indirect effect = 0.06, 95% CI = [0.03, 0.09]) were statistically significant. Similarly, for learner–learner interaction, the indirect effects through social presence (indirect effect = 0.09, 0.17) and learning engagement (indirect effect = 0.17, 95% CI = [0.11, 0.24]) were significant, as was the serial mediation through both mediators (indirect effect = 0.07, 95% CI = [0.04, 0.11]). (4) social presence and learning engagement played crucial mediating roles in the links between social interactions and online learning efficiency, and the predictive efficacy of learner–learner and learner–instructor interactions on online learning efficiency was found to be unequal.

## 1. Introduction

The integration of technology in education has revolutionized the way learning is facilitated and experienced, particularly with the advent of computer-assisted learning and educational technology tools. As remote instruction becomes increasingly commonplace, online learning has emerged as a compelling alternative to traditional classroom settings, without the barriers of time and distance [1], making the learning process more learner-centered [2,3]. This trend is especially pronounced at the graduate and undergraduate levels, and increasingly among middle school students [4,5], a demographic that is not only tech-savvy but is also at a critical stage of cognitive and social development [6]. With online learning efficiency as the ultimate goal of education [7,8], it is imperative for researchers, practitioners, and policymakers to prioritize the study and enhancement of learning efficiency. This focus will not only lead to improved academic achievements for students but also contribute to the broader goals of educational equity and access [9,10].

However, the mere presence of technology does not guarantee improved efficiency. It is the strategic integration of these tools within the learning process that can lead to significant gains in learning efficiency [11,12]. Increasingly, researchers ascribe high value to probing the significant factors, mainly individual factors and social factors, that may predict online learning efficiency and the underlying mechanisms [13,14,15,16]. Based on constructivist learning theory, learning is a process in which learners actively construct meaning in the process of social interaction with others on the basis of original knowledge and experience [17]. However, not all ages of learners can benefit from online learning. Middle students struggle with online learning because of their underdeveloped metacognitive skills, making it difficult to concentrate on online learning without teachers’ guidance; high school students prefer more peer interactions and recommendations for learning resources [18]. This means that the external social factors in the learning environment affect the learning efficiency through the internal individual factors of learners. In the context of online learning for middle school students, instructors and peers are significant members of the learning community, and the interactions between them and learners are concrete manifestations of social interaction, representing an important external social factor that affects learning efficiency. 

Recent advancements in educational technology have introduced sophisticated tools and platforms that facilitate interactive and personalized learning experiences, thereby enhancing learning efficiency [19]. The rapid integration of technology in educational practices has set the stage for a more nuanced exploration of the relationship between social interaction and learning efficiency within digital environments. It is of great practical significance to explore whether and how social interaction can promote online learning efficiency among middle school students. The results can help educators, policymakers, and educational technologists with developing strategies collaboratively to enhance the online learning experience, optimize social interaction, and improve learning efficiency in an increasingly digital world.

## 2. Literature Review

### 2.1. Social Interaction and Online Learning Efficiency

The importance of social interaction for learning has long been delineated by social constructivist theories, which are more focused on the effects of social interaction, language, and culture on learning and propose that cognitive growth is most strongly promoted through discussion, conflict, and collaboration [20,21]. In recent studies on educational technologies, researchers point out that research based on behavioral, systems, and cognitive theories of online interaction research simplify the complex interaction process in the context of online learning environments, while constructivism provides different forms of theoretical foundations for effective online learning environments [22,23]. For the above reasons, this study analyzes online interaction based on the theoretical framework of social constructivist theories. In addition, transactional distance theory [24] indicated that positive interaction plays a key role in reducing the potential misunderstanding caused by psychological and communicative distance in online learning. Social presence as online learners’ perceptions of others (e.g., peers) and projection of themselves as real in an online environment has been regarded as key to supporting successful online education experiences [25,26]. Therefore, this study also takes these two theories into the theoretical framework.

Social interaction, which mainly includes learner–learner interaction and learner–instructor interaction, has emerged as a critical component in online learning [17,27,28]. The most prominent framework of interaction in online education includes three major aspects: teacher–student interaction, student–student interaction, and student–content interaction [29]. The present study focuses on social (person–person) interaction; thus, student–content interaction is ignored below. Learner–learner interaction refers to two-way communication among learners, such as exchanging course content, discussing with each other and getting feedback from other learners; learner–instructor interaction refers to two-way communication between students and instructors [29,30], such as asking questions, providing support and giving encouragement for learners. Based on transactional distance theory [24], learners who actively engage in interaction are more likely to benefit from online learning [14,31,32]. 

Existing research predominantly focuses on the impact of interaction on online learning motivation [33,34,35,36], learning engagement [37,38,39,40] and academic emotions [41,42,43,44], learning performance [45,46,47,48,49,50], and elucidating the intricate dynamics between interaction and different aspects of online learning. Recent studies have oriented the focus of online interaction outcome variables towards learning efficiency; the role of the educator in fostering an efficient online learning environment cannot be overstated. Educators must be adept at facilitating meaningful interactions, providing timely feedback, and scaffolding learning experiences that are responsive to the diverse needs of their students [51]. Thi and Thuy [13] suggest that while instructor–student interaction is prevalent, the optimal interaction pattern for enhancing learning efficiency may vary based on the unique characteristics of each class. The research underscores that effective interaction patterns are likely those that stimulate active student engagement, facilitate communication between and among students, and support reflective practice in language use. Abuhmaid et al. [14] revealed that the presence of a supportive community further enhances the effectiveness of project-based learning in an online setting, while in-class students tend to have stronger views on project-based learning. The aforementioned study indicates that online interaction has a positive predictive effect on learning efficiency.

Although the direct relationship between social interaction and online learning efficiency has been explored, there are several important limitations. First, some researchers ignored the mutual effects of different types of social interactions; while the predicted extent on learning efficiency can vary with the participants involved [52], it is essential to account for both learner–learner and learner–instructor interactions in educational analyses. Second, existing research frequently draws on college student samples to examine the factors influencing online learning efficiency. As for middle school students, which often face issues such as sub-optimal learning outcomes [53], lack of self-control [54] and volatile academic emotions [55], the attention of researchers is also merited. Third, the individual variables (e.g., social presence and engagement) and proximal mechanisms that link social interaction to learning efficiency within the specific context of online learning environments are still under-explored [15] and warrant further investigation. To address these limitations, the present study aims to gain a deeper insight into the role of social presence in shaping students’ interactions, aiming to significantly enhance engagement and, consequently, the efficiency of online learning among middle school students.

### 2.2. Social Presence as the Mediator of the Relationship between Social Interaction and Learning Efficiency

Social presence encapsulates the perceived ‘realness’ of peers and instructors within the virtual classroom, serving as a cornerstone for a community of inquiry that is conducive to learning [56,57]. The enhancement of social presence through interaction is posited to engender a more profound sense of connectedness and belonging among learners, which in turn lubricates the gears of learning efficiency.

Interaction in online environments is characterized by a tapestry of discussions, collaborative endeavors, and peer-to-peer feedback, all of which are instrumental in nurturing an active and participatory learning milieu [58]. The interplay of these interactions is theorized to amplify the social presence, thereby creating a ripple effect on learning efficiency. Research shows that social interaction enhances social presence [59,60,61]. As highlighted in the study by Zhao et al. [62], interaction is the cornerstone of fostering social presence in online peer review groups, cultivating a sense of social presence through dynamic and responsive exchanges among participants and creating an environment conducive to collaborative learning. 

Students with higher levels of perceived social presence in an online environment experience a greater degree of learning satisfaction and efficiency [63]. A number of studies have found social presence to have an impact on student satisfaction [64,65]. Further research has indicated that the sense of social presence positively predicts learners’ perceptions of learning. Likewise, Cobb’s work on nursing education found that social presence was highly correlated to both student satisfaction and perceived learning [66]. Using multivariate regression, he found that social presence accounted for 44% of the variance in overall satisfaction and 36% of the variance in perceived learning. Similarly, Kang and Im [65] conducted multiple regression analyses to determine the factors in instructor–learner interaction that predicted learners’ perceived learning and satisfaction in online courses. 

Overall, effective interaction patterns not only facilitate communication but also enhance the feeling of being part of a cohesive group, thereby promoting a more engaging and productive online learning experience. Empirical evidence has consistently illuminated the mediating role of social presence. Studies have demonstrated that heightened social presence, catalyzed by rich interactions, correlates with improved student engagement, satisfaction, and more efficacious learning outcomes [16,56,67,68,69]. Based on prior theory and empirical studies referenced above, it is reasonable to infer that social presence may mediate the association between interaction and learning efficiency.

However, existing studies have predominantly focused on the social interaction of college students in online learning environments, with a relative lack of attention given to adolescents, despite the fact that they are also participants in the experience of online learning. Adolescents, characterized by unique developmental stages, cognitive growth, and social–emotional needs, may experience online learning environments differently [70]. Developmental considerations, such as the significant influence of peer relationships and the quest for social identity during this formative period [71], may alter the dynamics of social presence. Moreover, the transition in adolescents’ cognitive abilities from concrete to abstract reasoning [72] could mediate how they engage with online interactions, potentially impacting the efficiency of their learning process. The role of social and emotional learning is also pronounced in this age group, suggesting that the quality of social interactions within online platforms may have a differential effect on learning outcomes compared to adult learners [73]. Furthermore, despite adolescents’ generally high digital literacy, the translation of this skill into effective learning through social presence remains an open question [74,75]. Therefore, the educational implications of these factors underscore the importance of designing age-appropriate and socially responsive online learning environments for adolescents. Based on prior theory and empirical studies referenced above, it is reasonable to infer that social presence may mediate the association between interaction and learning efficiency and needs to be verified in online learning of middle school students.

### 2.3. Learning Engagement as the Mediator of the Relationship between Social Interaction and Learning Efficiency

Community of inquiry suggests that social and cognitive presence, facilitated through interaction, are foundational for creating a robust learning environment [57,76]. When learners are engaged in meaningful discourse and collaborative activities, they exhibit higher levels of intellectual and emotional investment, which is indicative of effective learning engagement [77]. This engagement is thought to mediate the efficiency of learning by promoting a deeper processing of information, enhancing motivation, and encouraging the use of strategic learning approaches that lead to better learning outcomes in a shorter time frame [14,78].

Social interaction during the online learning process can enhance the engagement of learners in their educational activities. Martin et al. found that teacher–student interaction is the most important among Moore’s three types of interaction. Teachers can improve student engagement and learning by providing a variety of communication channels, support, encouragement, and timely feedback [79]. Handa [80] evidenced the positive relationship between teacher–student interaction and learners’ affective engagement. It was established that verbally explicit immediate interaction influenced learners’ self-perceived cognitive and affective learning and, therefore, increased engagement in online class environment. Through student–student interaction, students attempted to reconcile what they learn with what they previously believed; they demonstrate growth in understanding, values and commitment typical of mature cognitive development [81]. Klisc et al. found that online discussion engages learners in critical thinking and more constructive learner–learner interaction due to their socially structured exchange of information [82], which implies greater cognitive engagement on the part of the learners.

Empirical evidence supports the notion that a strong sense of engagement, catalyzed by interaction, can significantly influence learning efficiency. Dabbagh and Kitsantas [16], for instance, have demonstrated a positive correlation between the level of engagement in online courses and improvements in learning efficiency. The mediating role of learning engagement is particularly important in online environments where the physical co-presence of learners and instructors is absent, and interaction must be intentionally designed to foster a sense of community and active participation [83]. Xu et al. [84] revealed that teacher facilitation could improve the level of learner–instructor interaction, thus increasing students’ engagement in online discussion. When instructors engage with students through personalized feedback, clear explanations, and cognitive challenges, students are likely to become more engaged with the learning material. This engagement, in turn, can lead to more efficient learning as students are motivated to invest effort and time in understanding and mastering the subject matter [85,86]. Lai et al. [87] found that more peer interaction in online learning community was positively associated with better students’ engagement. Peer interactions can enhance engagement by providing a variety of perspectives, promoting peer learning, and encouraging the co-construction of knowledge [88,89]. The social dynamics of these interactions can lead to increased motivation and a more efficient learning process as students work together to achieve common goals.

Based on prior theory and empirical studies referenced above, it is reasonable to infer that the mediating effect of learning engagement is a concept that holds promise for understanding the complex interplay between interaction and learning efficiency in educational settings.

### 2.4. Social Presence and Learning Engagement as the Mediator of the Relationship between Social Interaction and Learning Efficiency

Social presence, as conceptualized by Short et al. [90], is the perception of being with others in a mediated environment, which in turn fosters a sense of community and belonging. This sense of presence is pivotal as it is caused by interaction and enhances the quality of interaction, leading to increased learning engagement [76]. Learning engagement, characterized by the investment of time, effort, and interest in learning activities [73], is a critical factor that mediates the impact of interaction on learning efficiency. When learners are engaged, they are more likely to be motivated, to employ effective learning strategies, and to achieve learning objectives in a timely manner, thereby enhancing learning outcomes.

Community of inquiry suggests that social presence and learning engagement are interrelated and sequentially build upon each other to influence learning outcomes [76]. Empirical evidence supports the idea that interactions in online environments, which bolster social presence and can lead to higher levels of learning engagement and, consequently, more efficient learning [16,91]. Therefore, we inferred that social presence and learning engagement are the mediators of the relationship between interaction and learning efficiency. In addition, due to the social and cognitive development characteristics of adolescents, this effect is more prominent and should be verified preferentially in middle school students.

## 3. Research Questions

The conceptual model presented in this study is an adaptation of Moore’s online learning interaction model, which has been expanded to incorporate the potential mediating roles of online learning social presence and learning engagement. This enhancement aims to delve into the intricate dynamics between interaction and learning efficiency within the context of online learning environments. Drawing from existing theories and empirical evidence, our study addresses a gap in the literature by examining the multifaceted interplay among interpersonal interactions, social presence, learning engagement, emotions, and learning efficiency—a complex nexus that has received scant attention in prior research. As depicted in Figure 1, this adapted model underscores the significance of these mediating variables in understanding the efficacy of online learning interactions.

Specifically, this study examines the following: (1) whether social interaction will predict students’ learning efficiency; (2) whether social presence mediates the relationship between social interaction and learning efficiency; (3) whether learning engagement mediates the relationship between social interaction and learning efficiency; and (4) whether social presence and learning engagement exert serial mediating effect on the association between social interaction and learning efficiency.

The current study has the following hypotheses: 

**Hypothesis** **1.**
*Social interactions are actively related to online learning efficiency.*


**Hypothesis** **2.**
*Social presence will mediate the relationship between social interaction and learning efficiency.*


**Hypothesis** **3.**
*Learning engagement will mediate the association between social interaction and learning efficiency.*


**Hypothesis** **4.**
*Social interaction will predict learning efficiency through sequential mediating roles of social presence and learning engagement.*


## 4. Method

### 4.1. Participants and Procedure

This study conducted an in-class survey in the spring semester of 2023. A total of 371 students from 3 junior high schools in Hubei Province and Shanxi Province were selected as research participants. These three schools were forced to start teaching online during the COVID-19 pandemic and continue to do so when needed after the pandemic, compared to face-to-face teaching before the pandemic. The investigation protocol was approved by the administration of the school prior to data collection. First, the school administration invited class teachers to participate in this study. Second, class teachers informed and obtained the consent of parents, and then organized students to complete the questionnaire collectively. To protect the privacy of the participants and ensure the simplicity of the data collation and storage, each participant completed the questionnaire anonymously and was named with a numerical code after completing it. All data stored in encrypted electronic devices could only be read by members of the research team and only used for research purposes.

We eliminated data from participants who did not complete the entire questionnaire or who took an excessively long time, as well as data from students who provided invalid responses. A total of 344 valid questionnaires were obtained. The mean age of the remaining 344 participants was 13.61 years (SD = 0.73) with a range from 12 to 15 years old, and 195 (56.68%) of them were women. Before filling out the questionnaire, all participants were informed of the purpose of the investigation and participation was voluntary. Then, participants completed a series of questionnaires including demographic information and the items assessing online interaction, social presence, learning engagement and learning efficiency. All the questionnaires were in Chinese.

### 4.2. Measures

#### 4.2.1. Social Interaction

In this study, the measurement of social interaction was conducted utilizing the Online Learning Student Interaction Scale developed by Kuo et al. [92], with appropriate modifications tailored to the context of Chinese students. The instructor–student interaction sub-scale comprises six items, exemplified by the statement “I would inquire about issues to teachers through internet platforms such as QQ, WeChat, Tencent Classroom, etc.” The scale is scored on a 7-point Likert scale, with 1 indicating “strongly disagree” and 7 indicating “strongly agree”. The student–student interaction subscale consists of eight items, exemplified by the statement, “I would exchange course-related content with classmates through internet platforms such as QQ, WeChat, Tencent Classroom, etc.” It also employed a 7-point Likert scoring system, with the same interpretation of endpoints. The Cronbach’s alpha coefficients for the two sub-scales were 0.805 and 0.928, respectively, indicating high reliability.

#### 4.2.2. Social Presence

The measurement of social presence is based on the Social Presence Scale developed by Wei et al. [93], which encompasses three dimensions: co-presence, intimacy, and immediacy, each consisting of four items (including one reverse-scored item), totaling 12 items. The dimension of co-presence is exemplified by the item “In online classes, I feel a sense of being in the same place with others”. Intimacy is illustrated by the item “In online classes, I have a harmonious relationship with others”. The dimension of immediacy is captured by the item “In online classes, I find that I am respected by others”. The scale utilizes a 7-point Likert scoring system, where 1 represents “strongly disagree” and 7 represents “strongly agree”. The scale has demonstrated a high reliability coefficient of 0.879.

#### 4.2.3. Leaning Engagement

The measurement of student engagement in online learning was conducted using the Student Online Learning Engagement Scale developed by Dixson [94], which originally comprised four dimensions: skill engagement, affective engagement, behavioral engagement, and learning performance, with a total of 19 items and a reliability coefficient of 0.95. Considering that the behavioral engagement dimension primarily pertains to behaviors within online forums, which is not aligned with the research context, this study only employed the other three dimensions. Based on the factor loading values, a selection is made of four items for skill engagement and affective engagement, and two items for learning performance, totaling ten items. Skill engagement is exemplified by the item “Ensuring regular study”, affective engagement by “Applying course materials to my life”, and learning performance by “Performing well in exams or quizzes”. The scale uses a 7-point Likert scoring system, with 1 indicating “strongly disagree” and 7 indicating “strongly agree”. 

#### 4.2.4. Online Learning Efficiency

The measurement of learning efficiency is based on the Online Learning Efficiency Questionnaire developed by Tarafdar et al. [95], which includes four items, such as “Online learning platforms help me improve the quality of my learning”. The scale utilizes a 7-point Likert scoring system, where 1 indicates “strongly disagree” and 7 indicates “strongly agree”, and has demonstrated a high reliability coefficient of 0.911.

## 5. Results

### 5.1. Common Method Bias Test

In view of the fact that the data of all subjects in the current study were collected using self-rating scales, there may be common method bias, so the Harman single-factor method was used to test them. After the main analysis, eight characteristic factors were found to be 1. The results showed that the first common factor accounted for 38.44% of the variation, and the critical criterion was less than 40%, indicating that common method bias was not found in this study.

### 5.2. Descriptive Analysis and Correlations between Overall Variables

The basic descriptive data for interaction, social presence, learning engagement and learning efficiency are shown in Table 1. Pearson correlation analysis was used to test the bivariate correlations of all the variables. Table 1 shows that all the variables are significantly correlated with each other. Learner–learner interaction was significantly positively correlated with learner–instructor interaction, social presence, learning engagement and learning efficiency (*r* = 0.77, *p* < 0.01; *r* = 0.60, *p* < 0.01; *r* = 0.64, *p* < 0.01; *r* = 0.42, *p* < 0.01). Learner–instructor interaction was significantly positively correlated with social presence, learning engagement and learning efficiency (*r* = 0.60, *p* < 0.01; *r* = 0.68, *p* < 0.01; *r* = 0.44, *p* < 0.01). 

### 5.3. The Chain Mediation Model between Social Interactions and Online Learning Efficiency

After controlling for variables such as gender, age, and age of initial internet exposure, learner–learner interaction was found to significantly and positively predict learning efficiency (*β* = 0.40, *p*  <  0.001); the model demonstrates a good fit (R^2^ = 0.18, *p* < 0.001).

This study employs Model 6 of the PROCESS macro for SPSS 22.0, as articulated by Hayes (2013), to conduct a rigorous statistical analysis aimed at elucidating the mediating mechanisms within our theoretical framework. Figure 2 presents the results from the mediation of social presence and the learning engagement between learner–learner interaction and learning efficiency. In the first step, gender, age, and the age of initial internet exposure were included as control variables in the regression model; the regression analysis indicates a satisfactory model fit (R^2^ = 0.36, *p* < 0.001). Learner–learner interaction was found to have a significant and positive association with social presence (*β* = 0.50, *p*  <  0.001). In the second step, both learner–learner interaction and social presence were observed to show a significant and positive association with learning engagement (*β* = 0.36, *p*  <  0.001, *β* = 0.29, *p*  <  0.05); the regression analysis indicates a satisfactory model fit (R^2^ = 0.49, *p* < 0.001). In the third step, mediation analysis was performed to assess the association between learner–learner interaction, social presence, learning engagement and learning efficiency; the regression analysis indicates a good model fit (R^2^ = 0.29, *p* < 0.001). Only social presence and learning engagement were observed to show a significant and positive association with learning efficiency (*β* = 0.17, *p*  <  0.05, *β* = 0.48, *p*  <  0.001). 

The indirect effects of learner–learner interaction (indirect effect = 0.09, 95% CI = [0.00, 0.17]) on learning efficiency through social presence were significant. The indirect effects of learner–learner interaction (indirect effect = 0.17, 95% CI = [0.11, 0.24]) and learning efficiency through learning engagement were significant. The indirect effects of learner–learner interaction (indirect effect = 0.07, 95% CI = [0.04, 0.11]) on learning efficiency through social presence and learning engagement were significant. These results suggested that social presence and learning engagement serially mediated the relationship between learner–learner interactions and learning efficiency.

After controlling for variables such as gender, age, and age of initial internet exposure, learner–instructor interaction was found to significantly and positively predict learning efficiency (*β* = 0.43, *p*  <  0.001); the model demonstrates a good fit (R^2^ = 0.20, *p* < 0.001).

Model 6 of the PROCESS macro was used for mediation analysis. Figure 3 presents the results from the mediation of social presence and the learning engagement between learner–instructor interaction and learning efficiency. In the first step, gender, age, and the age of initial internet exposure were included as control variables in the regression model; learner–instructor interaction were found to have a significant and positive association with social presence (*β* = 0.60, *p*  <  0.001); the regression analysis indicates a good model fit (R^2^ = 0.36, *p* < 0.001). In the second step, both learner–instructor interaction and social presence were observed to show a significant and positive association with learning engagement (*β* = 0.52, *p*  <  0.001, *β* = 0.27, *p*  <  0.001); the regression analysis indicates a satisfactory model fit (R^2^ = 0.52, *p* < 0.001). In the third step, mediation analysis was performed to assess the association between learner–instructor interaction, social presence, learning engagement and learning efficiency; the regression analysis indicates a satisfactory model fit (R^2^ = 0.29, *p* < 0.001). Only social presence and learning engagement were observed to show a significant and positive association with learning efficiency (*β* = 0.14, *p*  <  0.05, *β* = 0.35, *p*  <  0.001).

The indirect effects of learner–instructor interaction (indirect effect = 0.08, 95% CI = [−0.00, 0.17]) on learning efficiency through social presence were not significant. The indirect effects of learner–instructor interaction (indirect effect = 0.18, 95% CI = [0.11, 0.26]) on learning efficiency through learning engagement were significant. The indirect effects of learner–instructor interaction (indirect effect = 0.06, 95% CI = [0.03, 0.09]) on learning efficiency through social presence and learning engagement were significant. These results suggested that social presence and learning engagement serially mediated the relationship between learner–instructor interactions and learning efficiency.

## 6. Discussion

This study, grounded in Social Presence Theory, Transactional Distance Theory, and Constructivist Learning Theory, explores the relationship between social interaction and online learning efficiency, as well as the underlying mechanisms. The findings indicate that the two types of social interactions (learner–learner and instructor–learner) are significantly and positively correlated with online learning efficiency. Regression analysis results also show that social interaction has a significant positive predictive effect on online learning efficiency, thus validating Research Hypothesis 1. However, when social presence and learning engagement are included in the regression equation, the direct predictive effect of social interaction on online learning efficiency becomes non-significant. The results of the mediation test indicate that social presence and learning engagement play a complete mediating role in the impact of social interaction on online learning efficiency. This mediating effect encompasses three pathways: the individual mediating effects of social presence and learning engagement, as well as their serial mediating effects. The mediating paths for instructor–learner interaction and learner–learner interaction are slightly different, leading to partial confirmation of Research Hypotheses 2–4.

First, this study found that social interaction positively predicts students’ learning efficiency during online learning. The result supports Hypothesis 1, which holds that all of two types of interactions are actively related to learning efficiency. Transactional Distance Theory (TDT), as articulated by Moore [24], underscores the pivotal role of interaction in the learning process. It posits that the interplay among distance education participants interactions can significantly influence the psychological and communicative distance of online education. By supporting Hypothesis 1, the study reinforces the theoretical underpinnings of TDT, which posits that the quality and intensity of interactions among online learners are critical for reducing transactional distance and enhancing learning efficiency. Empirical studies have consistently shown that learners who engage in high-quality and high-intensity interactions exhibit higher levels of learning efficiency, as evidenced by improved retention rates, faster knowledge acquisition, and more effective problem-solving skills [96,97]. These findings underscore the importance of creating an interactive learning environment that supports and enhances the learning process. In the realm of online education, the cognitive processes underpinning learning efficiency are significantly influenced by the strategic roles that educators play, with pivotal factors emerging as particularly influential: cognitive scaffolding, meta-cognitive development, and cognitive engagement [98]. In the domain of online learning, instructors employ cognitive scaffolding to facilitate the cognitive processing of complex information [99]. This pedagogical approach involves the systematic breakdown of intricate tasks into smaller, more digestible components, thereby enabling learners to construct knowledge in a stepwise fashion. By providing a structured framework, instructors guide learners through the learning process, ensuring that they can grasp foundational concepts before progressing to more advanced topics. Instructors not only mitigate cognitive overload but also fosters a deeper and more enduring understanding of the subject matter, which is paramount for efficient learning. Moreover, instructors play a pivotal role in nurturing higher-order thinking abilities by encouraging learners in self-assessment, goal setting, and strategic planning [100]. Meta-cognitive activities, such as reflection on learning strategies [101] and the evaluation of one’s cognitive processes [101], empower students to become autonomous learners. This proactive approach to learning in online environments bolsters the efficiency of cognitive processing and cultivates the self-directed learning and adaptability essential for navigating the dynamic and continuously evolving digital knowledge landscape [102]. Last but not least, instructors stimulate learning engagement through interactive and thought-provoking instructional methods that prompt learners to actively participate in the construction of knowledge. Techniques such as discussion forums, collaborative projects, case studies, and simulations serve to immerse learners in the subject matter, thereby promoting a deeper level of cognitive processing. When learners are deeply engaged in online learning, they are more likely to employ critical thinking and analytical skills, which are essential for the comprehensive understanding and retention of information [87]. Educators may further enhance cognitive engagement by posing open-ended questions, challenging assumptions, and encouraging creative and divergent thinking. This heightened level of engagement not only boosts learning efficiency but also fosters a genuine interest and lasting retention of the learning material. Certainly, there are two overarching points that encapsulate the reasons why online student–student interaction might not enhance learning efficiency: (1) despite the potential for collaboration, online student interactions often lack the immediacy of face-to-face communication, which research suggests is crucial for deep learning [83]. Without real-time engagement, students may not receive the prompt feedback necessary for efficient learning. (2) Studies indicate that the mere presence of interaction does not guarantee learning efficiency [103]. If student interactions are not well-facilitated or do not directly contribute to learning objectives, they may lead to a diversion from focused study, thus not enhancing learning efficiency. The study challenges the assumption that the mere presence of interaction is sufficient for learning efficiency, suggesting that the quality of interaction is paramount, thus enriching the theoretical understanding of interaction in online learning. Future research should concentrate on strategies to elevate the quality of interactions within online learning environments, as higher quality engagement has been linked to more efficient learning outcomes [104]. By identifying and implementing methods that foster meaningful, constructive dialogue among students, research can provide insights into how to optimize the educational benefits derived from online collaborative experiences.

Secondly, according to Hypothesis 2, social presence will mediate the relationship between the learner–learner interactions but not learner–instructor and online learning efficiency. The results are partly consistent with the prediction of Hypothesis 2 and support the social presence theory. Social presence theory posits that the degree to which individuals perceive each other as “real people” and feel connected during mediated communication is a critical factor in the effectiveness of that communication [56,105]. This perception of being in the company of others, even in a virtual environment, influences the social dynamics and the quality of interaction among participants. The application of social presence theory in the field of online learning enriches the educational experience by creating a sense of community and connection among students. This fosters interaction, collaboration, and motivation, ultimately contributing to a more engaging and successful online learning environment. However, it is interesting to note that social presence significantly and positively only predicts learning efficiency in online interactive contexts among students. From the student’s perspective, the enhancement of social presence in online learning is intricately linked to the quality of interactions. When students engage in interaction with peers, they can express and share ideas with others and gain a deeper construct of what they are learning, thus enhancing learning efficiency. Furthermore, support from others can also improve learners’ beliefs in exploring problems; the enhancement of intrinsic motivation also renders them more inclined to delve into the acquisition of knowledge [106]. As they engage with peers in collaborative projects and discussions, a shared learning journey emerges, reinforcing the feeling of being part of a cohesive group [107]. This camaraderie, underpinned by emotional connections and the development of communication skills, plays a critical role in establishing and maintaining social presence. Interestingly, the research findings suggest that the mediating effect of social presence on academic efficiency through learner–instructor interaction is not significant, which may be attributed to a variety of factors. Firstly, the online modality often falls short of replicating the dynamic and spontaneous nature of in-person exchanges. The limitations of asynchronous communication and the absence of non-verbal cues can lead to a superficial engagement that fails to stimulate the depth of interaction necessary for fostering a robust sense of social presence. Moreover, the asynchronous nature of online communication can create a lag in response times, which may hinder the natural flow of conversation and the establishment of a real-time dialogue. The immediacy of instructor feedback [108] is a crucial element in the learning experience, as it allows students to adjust their understanding and approach in a timely manner. When students receive personalized feedback from instructors, it affirms their individuality and contributes to a tangible sense of being seen and heard, which is fundamental to social presence [109]. When feedback is not promptly given, it can lead to a diminished sense of engagement and a reduced opportunity for students to feel acknowledged and comprehended by their instructors. This can be particularly problematic in the context of complex problem-solving or when students are grappling with new concepts that require immediate guidance. Instructor preparedness for online teaching is also a critical concern. Without adequate training in online pedagogical strategies, instructors may not effectively utilize the interactive tools available, leading to failure in creating an engaging and responsive learning atmosphere in line with the cognitive characteristics of adolescents that could enhance social presence and then enhance the learning efficiency.

Thirdly, learner–learner and learner–instructor interaction could predict higher learning efficiency through learning engagement. Hypothesis 3 is confirmed. Constructivist learning theory posits that learning is an active, constructive process deeply rooted in interaction and social participation [17]. By demonstrating the significance of social presence in fostering a sense of community and connection, the study enriches the theoretical understanding of how these factors contribute to a more engaging and successful online learning environment. The study extends the application of social presence theory to online learning contexts, highlighting its mediating role in the relationship between learner–learner interactions and learning efficiency. This reinforces the theory’s assertion that the perception of being with others in a virtual environment is crucial for effective communication and learning [56]. In the context of online learning, when students engage in discussions, collaborative projects, and peer feedback, they are engaged in a process that goes beyond merely sharing information; through these dynamic interactions, they are actively building and reinforcing their comprehension and knowledge base. This active involvement in the learning process fosters a deeper cognitive engagement with the material, as students are continually challenged to integrate new knowledge with their existing schemas. Consequently, as interactions among students and with instructors increase, so does the level of learning engagement, leading to a more meaningful and effective educational experience. The interactive constructivist environment thus nurtures a proactive approach to learning, where students are motivated to take ownership of their educational journey, directly enhancing their commitment and investment in the learning process. Online learning engagement, like its offline counterpart, enhances learning efficiency through active participation, clear goal setting, and the application of learned concepts. Students who are motivated and take an active role in their education, whether in a virtual classroom or a physical one, are more likely to achieve higher levels of comprehension and retention. However, online learning introduces unique elements. Online learning environments offer distinct advantages that bolster learning efficiency through enhanced flexibility and self-directed engagement. The capacity for learners to access educational materials at any time facilitates a self-paced learning modality, accommodating individual schedules and enabling learners to engage with content during their optimal periods of concentration. This represents a significant divergence from the structured timing inherent in traditional offline classes. Concurrently, the online milieu necessitates a heightened level of self-direction and self-motivation, as the immediacy of instructor and peer presence, which often catalyzes engagement in physical classrooms, is absent. This autonomy is empowering for learners who demonstrate discipline and initiative, allowing them to assume control of their educational trajectory. The convergence of these elements—flexibility, self-paced learning, self-direction, and self-motivation—underscores the potential of online learning to cater to individual needs and preferences, thereby fostering an efficient and effective learning experience.

Finally, learner–learner and learner–instructor interaction could predict learning efficiency through the sequential mediating effect of social presence and learning engagement. This result supports Hypothesis 4. As aforementioned, interactions were positively associated with learners’ social presence. Transactional Distance Theory (TDT) offers a pedagogical framework for understanding the nature of distance education. The study extends TDT by demonstrating how learner–learner and learner–instructor interactions can reduce transactional distance, thereby enhancing social presence. This provides a new pedagogical perspective on the role of interaction in online learning environments. At its core, TDT posits that distance education is not merely a function of geographic separation but is fundamentally a pedagogical concept. It describes the instructor–learner relationships that emerge when educators and students are separated by space or time. Learner–learner and learner–instructor interaction can reduce the transactional distance between instructors and students. When instructors and learners communicate frequently through online platforms, the immediate feedback and personalized support help to narrow the psychological and communication gap between them, thereby enhancing the students’ sense of social presence. Moore emphasized that dialogue is a key variable in reducing transactional distance. Interactions in online learning provide rich opportunities for dialogue, such as discussion forums, video conferencing, and real-time chats, all of which can strengthen communication between learner–learner or learner–instructor and foster the development of social presence. The finding coincides with previous research on online learning [110]. When courses are designed with a clear structure yet are adaptable to individual student needs, students are more likely to feel actively involved in the learning process, thus enhancing their sense of social presence and learning engagement. With increased interaction, students take on more responsibility in their learning process, which helps to enhance their autonomy. Autonomous learners often have higher motivation, which not only strengthens their sense of social presence but also promotes deeper learning engagement and higher learning efficiency. Therefore, social presence and learning engagement played a vital serial mediating role between social interaction and online learning efficiency. These results provide a new perspective of how different interactions were associated with learning efficiency in an online environment.

Several limitations of this study should also be noted. The cross-sectional methodology employed in our study provides a robust framework for assessing the relationships between social interactions and learning outcomes, offering valuable insights into the prevailing dynamics within the online educational sphere. Despite its limitations in capturing causality and temporal progression, this approach lays a solid foundation for hypothesis generation and preliminary exploration. Recognizing the need to elucidate the causal mechanisms and temporal dimensions, we advocate for the integration of longitudinal designs in subsequent research endeavors. Second, it is acknowledged that the model of interaction extends beyond human-to-human communication to encompass learner–content, learner–computer, and potentially learner–artificial intelligence interactions [111]. These additional dimensions of interaction hold significant potential for future research, as they may offer a more comprehensive understanding of the multifaceted nature of engagement in online learning environments. Third, this study focused solely on the interactions and outcomes among junior high school students in online learning, suggesting a need for future research to examine online learning dynamics across a wider range of age groups. Such exploration could yield insights into more effective approaches to enhance learning efficiency for diverse learners. Finally, considering that interaction may have different types, we only measured two traditional interactions (e.g., learner–learner and learner–instructor) but did not consider a more contemporary interaction model (e.g., learner–technology interaction, and learner–content interactions).

## 7. Implication

This research provides implications for students, instructors and administrators to improve the quality of online learning. Students are encouraged to proactively engage in online social interactions, leveraging their agency to enhance learning efficiency through active participation and self-directed learning strategies. By doing so, their online learning efficiency can be potentially improved. Learners should also be aware of the different forms of interaction available to them and leverage these to maximize their educational outcomes. From an instructor’s perspective, the findings suggest the importance of facilitating an interactive learning environment that promotes learner–learner and learner–instructor interactions. Educators should employ strategies that foster a sense of community and engagement, such as group projects, discussions, and real-time feedback sessions. Additionally, instructors are advised to recognize the unique contributions of both peer and instructor interactions to learning efficiency and balance these elements in their teaching approach. For educational administrators, the study’s results indicate the need to support the development of online platforms and curricula that integrate social interaction components effectively. In the spirit of constructivist philosophy, students are urged to embrace interactive experiences, actively constructing their knowledge through dynamic social engagement in the online learning environment. Administrators should consider policies and practices that encourage the use of collaborative tools and the creation of spaces for interaction. Moreover, they should advocate for research into the mechanisms of social presence and learning engagement to inform the design of more effective online learning experiences. In conclusion, the study’s implications advocate for a concerted effort from learners, instructors, and administrators to harness the power of social interactions in enhancing online learning efficiency. By understanding and applying these insights, each stakeholder can contribute to creating a more dynamic and effective online educational environment.

## Figures and Tables

**Figure 1 behavsci-14-00896-f001:**
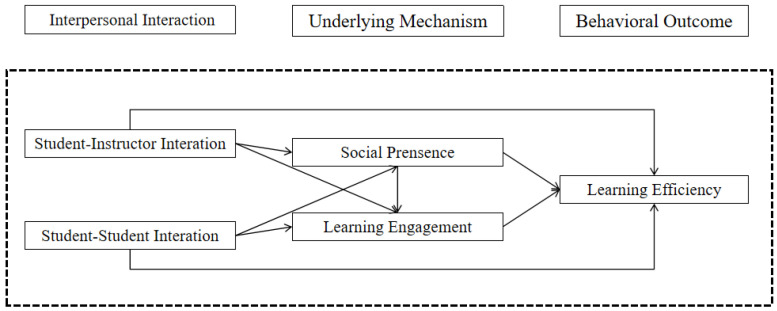
Online learning efficiency model.

**Figure 2 behavsci-14-00896-f002:**
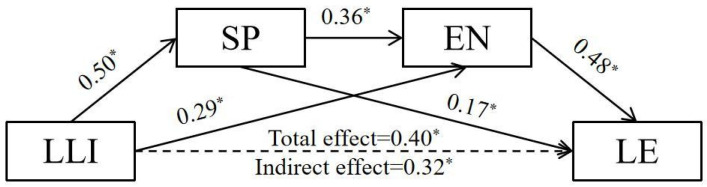
Mediating effect of SP and EN between LLI and LE. * *p* < 0.05.

**Figure 3 behavsci-14-00896-f003:**
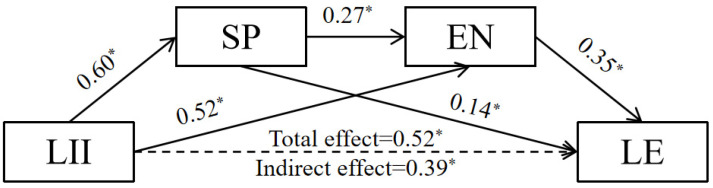
Mediating effect of SP and EN between LII and LE. * *p* < 0.05.

**Table 1 behavsci-14-00896-t001:** Descriptive analysis and correlations.

	M	SD	1	2	3	4	5
1. LLI	4.88	1.34	1				
2. LII	5.09	1.09	0.77 *	1			
3. SP	5.02	1.10	0.60 *	0.60 *	1		
4. EN	5.12	1.04	0.64 *	0.68 *	0.58 *	1	
5. LE	4.58	1.34	0.42 *	0.44 *	0.41 *	0.51 *	1

Note. LLI = learner–learner interaction; LII = learner–instructor interaction; SP = social presence; EN = engagement; LE = learning efficiency. * *p* < 0.05

## Data Availability

Data are available upon request.

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
