# Peer review of "Social Interaction and Online Learning Efficiency for Middle School Students: The Mediating Role of Social Presence and Learning Engagement"

_behavsci, 2024, doi:10.3390/bs14100896_

Round 1

Reviewer 1 Report

Comments and Suggestions for Authors

1. Abstract

The abstract plays a crucial role in summarizing the study's main findings. It is recommended that the abstract be revised to be more concise and direct, including key statistical data and relevant results. Emphasizing the central objective and the main conclusions can effectively capture the reader's attention, keeping them informed and engaged.

2. Theoretical Framework

The theoretical framework could be expanded to include more discussions on recent and relevant theories in the field of social interaction and online learning efficiency. Including comparisons with contemporary studies could provide a broader context and strengthen the arguments presented. Exploring authors such as Moore, Vygotsky, and Piaget, along with recent studies on educational technologies, can provide a more robust theoretical basis.

3. Ethical Issues (Authorizations)

Although the article mentions the ethical approval obtained, it is recommended to detail the informed consent procedures used, especially how the data of minor participants were protected. This may include descriptions of how confidentiality was maintained and how the data were stored and used by the institution's ethical standards and local laws.

4. Discussion and Conclusion

The discussion section could be enriched by including new sources that challenge or corroborate the results. A critical analysis of the study's limitations and suggestions for future research is also recommended. The conclusion should be expanded to include practical implications of the findings and how they might influence future educational practices and distance learning policies. Additionally, comparing the obtained data and previous studies can help better position the findings within the field of study.

5. General Text Review

A general text review is necessary to correct formatting inconsistencies, such as missing or excess spaces in various places. These adjustments will ensure the document has a visually coherent and professional presentation. The review should also focus on language uniformity and correcting any grammatical or spelling errors, ensuring the content is clear and accessible to the reader.

Comments on the Quality of English Language

Good

Author Response

We greatly appreciate the comprehensive and constructive feedback provided. In response to your specific suggestions, we have made the following revisions.

Comments 1: The abstract plays a crucial role in summarizing the study's main findings. It is recommended that the abstract be revised to be more concise and direct, including key statistical data and relevant results. Emphasizing the central objective and the main conclusions can effectively capture the reader's attention, keeping them informed and engaged.

Response 1: We have refined the abstract to be more succinct, incorporating key statistical data and study outcomes. The central objectives and conclusions are now emphasized for better clarity and engagement.

The new abstract is as follows:

Abstract: (1) Background: learning efficiency is recognized as the ultimate goal of online education, as it is related to the quality of online education and the cognitive development of students, and is influenced by social interactions. This study explore the mediating roles of social presence and learning engagement in the relationship between social interaction and online learning efficiency, to contribute to gaps remain in identifying the underlying mechanisms of prior studies. (2) Methods: students from three middle schools(N=344; Mage = 13.61; 56.68% women) completed self-report questionnaires regarding social interaction, social presence, learning engagement, and learning efficiency. (3) Results: the study findings reveal significant serial mediation effects of social presence and learning engagement in the relationship between learner-instructor and learner-learner interactions and learning efficiency. Specifically, while the indirect effect of learner-instructor interaction through social presence alone(indirect effect = 0.08, 95% CI =[-0.00, 0.17]) was not significant, the pathways through learning engagement(indirect effect = 0.18, 95% CI = [0.11, 0.26]) and the combined mediation through both social presence and learning engagement(indirect effect = 0.06, 95% CI =[0.03, 0.09]) were statistically significant. Similarly, for learner-learner interaction, the indirect effects through social presence(indirect effect = 0.09, 0.17) and learning engagement(indirect effect = 0.17, 95% CI = [0.11, 0.24]) were significant, as was the serial mediation through both mediators(indirect effect = 0.07, 95% CI = [0.04, 0.11]). (4) Conclusions: social presence and learning engagement played crucial mediating roles in the links between social interactions and online learning efficiency, and the predictive efficacy of learner-learner and learner-instructor interactions on online learning efficiency was found to be inequality.

Comments 2: The theoretical framework could be expanded to include more discussions on recent and relevant theories in the field of social interaction and online learning efficiency. Including comparisons with contemporary studies could provide a broader context and strengthen the arguments presented. Exploring authors such as Moore, Vygotsky, and Piaget, along with recent studies on educational technologies, can provide a more robust theoretical basis.

Response 2: The theoretical framework has been enriched with discussions on contemporary theories related to social interaction and online learning efficiency. We have included comparative analyses with recent studies to strengthen our arguments and referenced seminal authors such as Moore, Vygotsky, and Piaget, along with contemporary research on educational technologies.

In the review section, we have made the following changes:

The importance of social interaction for learning has long been delineated by social constructivist theories, which are more focused on the effects of social interaction, language, and culture on learning, and propose that cognitive growth is most strongly promoted through discussion, conflict, and collaboration[20,21]. In recent studies on educational technologies, researchers point out that research based on behavioral, systems and cognitive theories of online interaction research simplify the complex interaction process in the context of online learning environments, while constructivism provides different forms of theoretical foundations for effective online learning environments[22,23]. For the above reasons, this study analyzes online interaction based on the theoretical framework of social constructivist theories. In addition, transactional distance theory[24] indicated that positive interaction plays a key role in reducing the potential misunderstanding caused by psychological and communicative distance in online learning. Social presence as online learners’ perceptions of others (e.g., peers) and projection of themselves as real in an online environment, has been regarded as key to supporting successful online education experiences[25,26]. Therefore, this study also takes these two theories into the theoretical framework.

In the research questions section, we have constructed a new model of online learning efficiency based on the theories of predecessors:

The conceptual model presented in this study is an adaptation of Moore's online learning interactions model, which has been expanded to incorporate the potential mediating roles of online learning social presence and learning engagement. This enhancement aims to delve into the intricate dynamics between interaction and learning efficiency within the context of online learning environments. Drawing from existing theories and empirical evidence, our study addresses a gap in the literature by examining the multifaceted interplay among interpersonal interactions, social presence, learning engagement, emotions, and learning efficiency—a complex nexus that has received scant attention in prior research. As depicted in Fig. 1, this adapted model underscores the significance of these mediating variables in understanding the efficacy of online learning interactions.

Fig. 1. Online learning efficiency model

Comments 3: Although the article mentions the ethical approval obtained, it is recommended to detail the informed consent procedures used, especially how the data of minor participants were protected. This may include descriptions of how confidentiality was maintained and how the data were stored and used by the institution's ethical standards and local laws.

Response 3: We have detailed the informed consent procedures, particularly regarding the protection of data for minor participants. This includes descriptions of confidentiality maintenance and adherence to ethical standards and school standard for data storage and usage. 

In the participants and procedure section, we have elaborated with enhanced precision and detail to fortify the academic rigor of our exposition:

This study conducted an in-class survey in the spring semester of 2023. A total of 371 students from three junior high schools in Hubei Province and Shanxi Province were as research participants. These three schools were forced to start teaching online during the COVID-19 pandemic and continue to do so when needed after the pandemic, compared to face-to-face teaching before the pandemic. The investigation protocol was approved by the administration of the school prior to data collection. First, the school administration invited class teachers to participate in this study. Second, class teachers informed and obtained the consent of parents, and then organize students to complete the questionnaire collectively. To protect the privacy of the participants and to ensure the simplicity of the data collation and storage, each participant completed the questionnaire anonymously and was named with a numerical code after completing it. All data stored in encrypted electronic devices that can only be read by members of the research team and only used for research purposes.

Comments 4: The discussion section could be enriched by including new sources that challenge or corroborate the results. A critical analysis of the study's limitations and suggestions for future research is also recommended. The conclusion should be expanded to include practical implications of the findings and how they might influence future educational practices and distance learning policies. Additionally, comparing the obtained data and previous studies can help better position the findings within the field of study.

Response 4: The discussion section now incorporates new sources that either challenge or support our findings. We have added a critical analysis of our study's limitations and proposed directions for future research. The conclusion has been expanded to discuss practical implications and potential influences on educational practices and policies, with comparisons to previous studies for better positioning within the field.

We have augmented the section on the application of our findings with additional substance to elucidate the practical implications of our research:

This research provides implications for students,instructors and administrators to improve the quality of online learning. Students are encouraged to proactively engage in online social interactions, leveraging their agency to enhance learning efficiency through active participation and self-directed learning strategies. By doing so, their online learning efficiency can be potentially improved. Learners should also be aware of the different forms of interaction available to them and leverage these to maximize their educational outcomes. From an instructor's perspective, the findings suggest the importance of facilitating an interactive learning environment that promotes learner-learner and learner-instructor interactions. Educators should employ strategies that foster a sense of community and engagement, such as group projects, discussions, and real-time feedback sessions. Additionally, instructors are advised to recognize the unique contributions of both peer and instructor interactions to learning efficiency and to balance these elements in their teaching approach. For educational administrators, the study's results indicate the need to support the development of online platforms and curricula that integrate social interaction components effectively. In the spirit of constructivist philosophy, students are urged to embrace interactive experiences, actively constructing their knowledge through dynamic social engagement in the online learning environment. Administrators should consider policies and practices that encourage the use of collaborative tools and the creation of spaces for interaction. Moreover, they should advocate for research into the mechanisms of social presence and learning engagement to inform the design of more effective online learning experiences. In conclusion, the study's implications advocate for a concerted effort from learners, instructors, and administrators to harness the power of social interactions in enhancing online learning efficiency. By understanding and applying these insights, each stakeholder can contribute to creating a more dynamic and effective online educational environment.

Comments 5: A general text review is necessary to correct formatting inconsistencies, such as missing or excess spaces in various places. These adjustments will ensure the document has a visually coherent and professional presentation. The review should also focus on language uniformity and correcting any grammatical or spelling errors, ensuring the content is clear and accessible to the reader.

Response 5:A thorough text review has been conducted to correct formatting inconsistencies and ensure visual coherence and professionalism. Language uniformity and grammatical accuracy have been addressed to enhance clarity and readability.

Reviewer 2 Report

Comments and Suggestions for Authors

The article explores an interesting topic in a population that has received little attention in relation to their online learning experiences. However, the authors do not characterise the type of learning experiences that this particular group of individuals have had. This information should be included.

While the authors acknowledge that the use of self-report measures is a limitation, it is important that they explain the rationale for conducting this type of research and that they better document previous uses and results of the research tools. 

Author Response

Thank you for your insightful feedback on our manuscript. We have taken your comments to heart and have made the following revisions.

Comments 1:The article explores an interesting topic in a population that has received little attention in relation to their online learning experiences. However, the authors do not characterise the type of learning experiences that this particular group of individuals have had. This information should be included.

Response 1:We acknowledge the oversight in not detailing the specific type of learning experiences of the population under study. To address this gap, we have now included a section that describes the nature of their online learning experiences, providing a clearer context for our research findings. In the participants and procedure section, we have elaborated with enhanced precision and detail:

These three schools were forced to start teaching online during the COVID-19 pandemic and continue to do so when needed after the pandemic, compared to face-to-face teaching before the pandemic. The investigation protocol was approved by the administration of the school prior to data collection.

Comments 2:While the authors acknowledge that the use of self-report measures is a limitation, it is important that they explain the rationale for conducting this type of research and that they better document previous uses and results of the research tools. 

Response 2: We understand the importance of justifying our choice of self-report measures. Additionally, we have rewritten the limitation section.

Reviewer 3 Report

Comments and Suggestions for Authors

Thank you for the opportunity to review the manuscript about social interactions and online learning efficacy. Even though I applaud the authors to their large sample size of school students, I have major concerns about the quality of the study, particularly its analyses.

The authors conducted their analyses on one questionnaire only – resulting in a correlative study. Which is fine, but I think the authors then need to be very careful with their choice of analyses and their interpretations. I advise to not use mediation analyses, since mediation analyses are based on causal assumptions instead of correlative relations. Thus, I advise the authors to stick to their choice of design and to appropriated analyses which are only of correlative nature.

In this vein, I struggled with understanding the presentation of the mediation analyses. I would stick to the APA guidelines and present the statistics of all three models and presenting the corresponding coefficients of models with a significant indirect effect within Figures. Thus, I would delete Table 2 and 4. I also would delete Table 3 and 5 and include these statistics within the text. And then present significant models in Figures with all coefficients (also the total effect and the direct effect, with significance (*) if appropriate).

Additionally, the authors chose to assess students’ self-reported learning efficacy which limits the interpretation of the results. The authors discussed this issue in the discussion section, but it also should be noted throughout the manuscript.

Did the authors preregister their analyses? This is essential to judge the quality and transparency of the study.

Please describe the design when describing the participants: correlative survey.

Please include information of the kind of reliability method in all sections – even if there are the same (Cronbach’s alpha).

Please check the manuscript in regard of spelling, grammatical errors, format errors (e.g., too many / too few spaces), as I observed many errors. I advise proof-reading.

Also, please check the APA guidelines and the guidelines of the journal and keep the headings consistent (capital letter versus lower case letters).

Table 1: The authors only described the meaning of two **p but not of *p or ***p like represented in the table. I recommend setting alpha-level to .05 and only refer to the p value as significant when it is below 0.05 and not to report *p or ***p. Also: In this table no explanations of LLI etc. are needed since there are no abbreviations.

Why did the authors control for gender, age, and the age of initial interest? Please explain. Are the results the same without controlling for these variables?

Please add effect sizes to all results.

Finally, I advise the authors to include more recent literature.

Comments on the Quality of English Language

I advise proof-reading (see comments above).

Author Response

Thank you for your insightful feedback on our manuscript. We have taken your comments to heart and have made the following revisions:

Comments 1: The authors conducted their analyses on one questionnaire only – resulting in a correlative study. Which is fine, but I think the authors then need to be very careful with their choice of analyses and their interpretations. I advise to not use mediation analyses, since mediation analyses are based on causal assumptions instead of correlative relations. Thus, I advise the authors to stick to their choice of design and to appropriated analyses which are only of correlative nature.

Please describe the design when describing the participants: correlative survey.

Response 1: "In this study, we propose a chained mediation model to explore the causal mechanisms linking social interaction to online learning efficiency. We delineate a clear sequence where social interaction first influences social presence, which in turn affects learning engagement, culminating in the enhancement of learning efficiency. Besides, social presence and learning engagement serially mediate the impact of social interaction on learning outcomes. By emphasizing these causal linkages, our model provides a robust framework for understanding the mediating processes in online educational settings. We are appreciative of the opportunity to strengthen our manuscript and are receptive to any additional feedback.

In the research questions section, we have constructed a new model of online learning efficiency based on the theories of predecessors:

The conceptual model presented in this study is an adaptation of Moore's online learning interactions model, which has been expanded to incorporate the potential mediating roles of online learning social presence and learning engagement. This enhancement aims to delve into the intricate dynamics between interaction and learning efficiency within the context of online learning environments. Drawing from existing theories and empirical evidence, our study addresses a gap in the literature by examining the multifaceted interplay among interpersonal interactions, social presence, learning engagement, emotions, and learning efficiency—a complex nexus that has received scant attention in prior research. As depicted in Fig. 1, this adapted model underscores the significance of these mediating variables in understanding the efficacy of online learning interactions.

Fig. 1. Online learning efficiency model

Comments 2: In this vein, I struggled with understanding the presentation of the mediation analyses. I would stick to the APA guidelines and present the statistics of all three models and presenting the corresponding coefficients of models with a significant indirect effect within Figures. Thus, I would delete Table 2 and 4. I also would delete Table 3 and 5 and include these statistics within the text. And then present significant models in Figures with all coefficients(also the total effect and the direct effect, with significance(*) if appropriate).

Response 2: We have adhered to the APA guidelines for presenting statistical models and have revised the presentation of our results. Tables 2 and 4 have been removed, and the corresponding statistics have been integrated narratively. We have also prepared figures to illustrate significant models with all coefficients, including total and direct effects, marked with significance indicators where appropriate. The entire section of the data results has been revised for accuracy and clarity.

For a detailed examination of the causal mechanisms and the empirical support for our chained mediation model, please refer to the 'Results' section(Section 5) of our paper. There, we present a thorough analysis of the data that substantiate the sequential mediating roles of social presence and learning engagement between social interaction and online learning efficiency.

Comments 3: Additionally, the authors chose to assess students’ self-reported learning efficacy which limits the interpretation of the results. The authors discussed this issue in the discussion section, but it also should be noted throughout the manuscript.

Response 3: We have emphasized the limitations of using self-reported measures throughout the manuscript and have noted the implications for the interpretation of our results. Additionally, we have rewritten the limitation section.

Several limitations of this study should also be noted. The cross-sectional methodology employed in our study provides a robust framework for assessing the relationships between social interactions and learning outcomes, offering valuable insights into the prevailing dynamics within the online educational sphere. Despite its limitations in capturing causality and temporal progression, this approach lays a solid foundation for hypothesis generation and preliminary exploration. Recognizing the need to elucidate the causal mechanisms and temporal dimensions, we advocate for the integration of longitudinal designs in subsequent research endeavors. Second, it is acknowledged that the model of interaction extends beyond human-to-human communication to encompass learner-content, learner-computer, and potentially learner-artificial intelligence interactions[111]. These additional dimensions of interaction hold significant potential for future research, as they may offer a more comprehensive understanding of the multifaceted nature of engagement in online learning environments. Third,this study focused solely on the interactions and outcomes among junior high school students in online learning, suggest a need for future research to examine online learning dynamics across a wider range of age groups. Such exploration could yield insights into more effective approaches to enhance learning efficiency for diverse learners.Finally, considering that interaction may have different types, we only measured two traditional interactions(e.g., learner-learner and learner-instructor) but did not consider a more contemporary interaction model(e.g., learner-technology interaction, and learner-content interactions).

Comments 4: Did the authors preregister their analyses? This is essential to judge the quality and transparency of the study.

Response 4: Thank you for your question regarding the preregistration of our study. While our research was not preregistered in the traditional sense, we adhered to a rigorous ethical review process and obtained approval from our institution's review board. This process ensured that our study design, methodology, and data collection procedures were thoroughly vetted and met the highest ethical standards. Although preregistration was not part of our initial research protocol, we understand the value it brings to research transparency and reproducibility. Our study's methodology and analysis plan were clearly defined prior to data collection and were consistent with the ethical approval guidelines.

Comments 5: Please include information of the kind of reliability method in all sections – even if there are the same (Cronbach’s alpha).

Response 5: Information regarding the reliability method, specifically Cronbach's alpha, has been included in all relevant sections(4.2 Measures).

Comments 6: Please check the manuscript in regard of spelling, grammatical errors, format errors (e.g., too many / too few spaces), as I observed many errors. I advise proof-reading. Also, please check the APA guidelines and the guidelines of the journal and keep the headings consistent (capital letter versus lower case letters).Table 1: The authors only described the meaning of two **p but not of *p or ***p like represented in the table. I recommend setting alpha-level to .05 and only refer to the p value as significant when it is below 0.05 and not to report *p or ***p. Also: In this table no explanations of LLI etc. are needed since there are no abbreviations.

Response 6: The manuscript has undergone thorough proofreading to correct spelling, grammatical, and formatting errors. We have also ensured consistency with APA guidelines and journal formatting requirements. We have updated Table 1 to adhere to an alpha-level of .05.

Comments 7: Why did the authors control for gender, age, and the age of initial interest? Please explain. Are the results the same without controlling for these variables?

Response 7: Referencing previous studies, we have incorporated control variables related to online learning into our data analysis. The correlational analysis results from our study found that age is significantly and negatively correlated with social interaction, social presence, learning engagement, and learning efficiency.

(Reference: Wang, Y., Cao, Y., Gong, S., Wang, Z., Li, N., & Ai, L. Interaction and learning engagement in online learning: The mediating roles of online learning self-efficacy and academic emotions. Learning and Individual Differences 2022, 94, 102128.)

Comments 8: Please add effect sizes to all results.

Response 8: Effect sizes have been added to all reported results to provide a more comprehensive understanding of the magnitude of our findings.

Comments 9: Finally, I advise the authors to include more recent literature.

Response 9: Our literature review has been updated with the most current studies, ensuring that our research is contextualized within the latest academic discourse.